# Effects of Er: YAG laser and acid etching on bond strength of clear aligner attachments to fluorotic enamel

Rui Xia[1], Jie Lei[1,2], Maoxuan Luo[1], Yao Xiao [1,2,3]*

1 Luzhou Key Laboratory of Oral & Maxillofacial Reconstruction and Regeneration, The Affiliated Stomatology Hospital, Southwest Medical University, Luzhou, Sichuan, China, 2 Department of Chengbei Outpatient, The Affiliated Stomatology Hospital, Southwest Medical University, Luzhou, Sichuan, China, 3 Department of Orthodontics, The Affiliated Stomatology Hospital, Southwest Medical University, Luzhou, Sichuan, China

☯ These authors contributed equally to this work.

* orthoxiaoyao@outlook.com

## Abstract

### Background

There is a lack of research on the bonding ability of attachments to fluorosis enamel. This study evaluates Er: YAG laser-assisted acid etching as a potential optimization protocol.

### Methods

Twenty healthy teeth and ninety fluorotic teeth (Thylstrup-Fejerskov Index = 4) were divided into a control group (healthy enamel + acid etching) and a fluorotic group (acid etching 30/60/90 seconds vs. Er: YAG laser + acid etching). The enamel surface was analyzed using scanning electron microscopy, and the shear bond strength (SBS) of each group ($n$ = 10/group) was tested.

### Results

Attachments exhibited higher SBS than brackets ($P$ < 0.01). Laser-acid etching enhanced SBS compared to acid etching alone ($P$ < 0.01). Laser-treated surfaces exhibited predominantly mixed fracture and resin cohesion fractures, in contrast to adhesive interface fractures and mixed fractures in the acid-only groups.

### Conclusion

Er: YAG laser (100mJ/30 Hz) with 60 seconds acid etching achieves excellent bonding for fluorotic enamel attachments, restoring adhesion to healthy enamel levels while preventing over-etching damage. This protocol shows clinical potential for bonding fluorotic enamel.

**Data availability statement:** All relevant data are within the paper and its Supporting information files.

**Funding:** The authors acknowledge financial support from the Southwest Medical University Young Scholars (2021ZKQN038), the Southwest Medical University Applied Basic Research General Program (2021ZKMS015). The funders had no role in study design, data collection and analysis, decision to publish, or preparation of the manuscript. There are no financial conflicts of interest to disclose.

**Competing interests:** The authors have declared that no competing interests exist.

**Abbreviations:** SBS, Shear Bond Strength; TFI, Thylstrup-Fejerskov Index; SEM, Scanning Electron Microscope.

## 1. Introduction

Dental fluorosis is a type of enamel developmental disorder caused by exposure to high concentrations of fluoride during dental development. It is pathologically characterized by a disturbance in the structure or arrangement of the enamel crystals [1]. Dental fluorosis has become one of the most significant diseases to prevent and control in China at present [2]. Recent studies suggest an increasingly high incidence of dental fluorosis globally, ranging from Mexico (15.5 to 100%), Nigeria (41.7%), and the United States (65%) [3–5]. Moreover, the demand for orthodontic treatment is constantly rising [6].

However, managing fluorotic enamel poses unique challenges in orthodontic practice, particularly during bracket bonding procedures [7,8]. The distinctive surface structure of dental fluorosis significantly reduces the bonding strength between enamel and brackets. Compared to healthy enamel, previous studies [9] demonstrated a decrease in shear bonding strength (SBS) between brackets and fluorotic enamel. This impaired bonding strength often results in bracket detachment, difficulties in bonding, prolonged treatment duration, and increased clinical costs. Conventional acid etching techniques, while effective on normal enamel, are inadequate for fluorotic surfaces due to their altered chemical composition and porosity characteristics. Therefore, this study aimed to investigate whether combining Er: YAG laser with acid etching could enhance the bonding strength of clear aligner attachments on fluorotic enamel compared to conventional acid etching alone.

Emerging evidence suggests Er: YAG laser technology presents a promising solution for enhancing fluorotic enamel bonding. Some researchers found that Er: YAG laser etching increased the bonding ability of fluorotic enamel by improving bonding area and providing mechanical bonding pattern [10]. The Er: YAG laser has been confirmed to enhance the structure and function of dental fluorosis, potentially alleviating poor bonding capability [11,12]. Furthermore, the Er: YAG laser is considered effective for re-bonding orthodontic brackets, providing a superior bonding approach [13,14]. Therefore, Er: YAG may enhance the reduction in bonding strength caused by the compromised fluorotic structure.

The popularity of clear aligners has led to a significant increase in demand for orthodontic treatment among adults. Clear aligners are preferred for their superior comfort and aesthetics compared to conventional fixed appliances. Aligner attachments consist of composite resin bonded to the tooth surface. However, the bonding strength between dental fluorosis and the attachment remains unknown. Reviewing the literature, there has been almost no research done on the bonding performance between fluorotic enamel and clear aligner attachments. Clear aligner attachments may provide an effective solution for dental fluorosis, ensuring a lower shedding ratio.

## 2. Materials and methods

### 2.1. Sample collection

This study was approved by the Biomedical Ethics Committee of the Affiliated Stomatology Hospital of Southwest Medical University (Lot No. 20201126001).

Written informed consent was obtained from all participants who signed the consent forms. The extracted premolars were collected from Gulin and Xuyong Counties in Luzhou City (a high incidence area of dental fluorosis) from October 2020 to December 2021. Evaluation of the fluorosis level is performed according to the Thylstrup-Fejerskov Index (TFI) [15], which classifies dental fluorosis in terms of its absence (TFI 0) through the presence of opaque lesions (TFI=3) that blend to overtake the entire surface of the enamel, producing the appearance of white chalk (TFI=4). In more advanced stages of fluorosis, there is a gradual loss of enamel and anatomical dental deformities (TFI=5–9). In this study, 20 healthy teeth (TFI=0) and 90 dental fluorosis (TFI=4) were randomly assigned to the control group (group O) and experimental groups (groups A, B, and C) using computer-generated random numbers, according to Table 1. Then the selected teeth were cleaned to remove periodontal tissue and stored in 1% chloramine solution at 4°C. The solution was renewed weekly to maintain tissue integrity. The maximum storage duration was 3 months.

Dental exclusion criteria: incomplete buccal surfaces of teeth, teeth with decay, restorations, chips, cracks, and dysplasia.

## 2.2. Sample preparation

**2.2.1. *Cleaning the dental surface*.** The teeth were cleaned for 5 seconds using a low-speed dental handpiece, rubber cup, and fluoride-free polishing paste, then rinsed for 5 seconds with water. The teeth were dried with clean air (oil-free and water-free) for 3 seconds.

**2.2.2. Er: YAG laser etching.** The tooth surface was uniformly etched with the Er: YAG laser according to the mode that targets the hard tissue of the tooth based on the device's instructions (Syneron, Israel) (2940nm, 100mJ energy, 30 Hz frequency, pulse duration 250 µs, 6/8 water volume, 1.3*14 mm working tip, 2 mm distance from the working tip to the dental surface, irradiating 10s). Uniform moving irradiation in the same way as stacked tiles, then the etched enamel surfaces were rinsed with water for 10 seconds and dried with clean air (oil-free and water-free) for 3 seconds.

**2.2.3. Acid etching.** The enamel samples in each group were acid etched using 37% phosphoric acid for 30 seconds (Heraeus Kulze, Germany), then rinsed with running water for 10 seconds and dried with clean air (oil-free and water-free) for 3 seconds (the etching time is shown in Table 1).

**2.2.4. Bonding bracket.** According to the Grengloo adhesive instructions (Ormco, USA), the brackets (floor area 11.80mm², Victory Series Standard Metal Bracket, 3M ESPE Dental Products, USA) for the maxillary premolar were bonded to the center of the clinical crown by the same orthodontist to ensure that the brackets were adhered to the same position of the tooth. Apply a very thin coat of Ortho Solo to the prepared tooth, and extrude a small amount of Grengloo adhesive paste onto the bracket pad. The bracket was placed at the center of the crown and positioned with gentle

**Table 1. Groups in the study.**

| Groups | | Interventions | *n* |
|---|---|---|---|
| O | O1 | Healthy teeth+ Acid etching 30s+Bracket | 10 |
| | O2 | Healthy teeth+ Acid etching 30s+ Attachment | 10 |
| A | A1 | Dental fluorosis+Acid etching 30s+ Bracket | 10 |
| | A2 | Dental fluorosis+Acid etching 60s+ Bracket | 10 |
| | A3 | Dental fluorosis+Acid etching 90s+ Bracket | 10 |
| B | B1 | Dental fluorosis+Acid etching 30s+ Attachment | 10 |
| | B2 | Dental fluorosis+Acid etching 60s+ Attachment | 10 |
| | B3 | Dental fluorosis+Acid etching 90s+ Attachment | 10 |
| C | C1 | Dental fluorosis+Er: YAG laser etching+Acid etching 30s+ Attachment | 10 |
| | C2 | Dental fluorosis+Er: YAG laser etching+Acid etching 60s+ Attachment | 10 |
| | C3 | Dental fluorosis+Er: YAG laser etching+Acid etching 90s+ Attachment | 10 |

pressure. After removing the excess adhesive, each bracket was irradiated for 10 seconds with a light-curing lamp (light curing machine LED. F, Woodpecker, China) at a distance of 1 mm from the tooth surface.

**2.2.5. Bonding attachment.** After tooth preparation (etch, rinse, and dry), the adhesive (Nano-technology Dental Adhesive, Dentsply, USA) was evenly applied on the enamel surface. The solvent was removed using a clean air-syringe, and the area was irradiated with a light-curing lamp for 5s. A 3 mm long vertical rectangular resin attachment (Z250 composite resin, Filtek Z250 restoration, ESPE Dental Products, 3M, USA) was chosen and securely filled within the template (Clear aligner, Invisalign, China). The attachment template used was unified for all attachments by employing an Invisalign attachment template of premolars with a rectangular attachment sized 3*2 mm, which was utilized to bond all the attachments. Subsequently, the attachment template was pressed firmly on the center of the tooth surface and illuminated with a light-curing lamp for 10 seconds. Afterward, the template was removed, and the excess resin was cleared away (the area at the bottom of the attachment was 6.00 mm$^2$). To clarify, Clear aligners were not particularly different from other invisible aligners used in clinical practice.

## 2.3. Observation of enamel surface structure by Scanning Electron Microscope (SEM)

One untreated healthy tooth, one untreated tooth with dental fluorosis, and a total of six teeth from group B1-C3 were randomly selected to prepare 5*5*2 mm specimens for SEM analysis. After drying for 24 hours, the buccal surfaces of the sectioned samples were coated with a gold-palladium layer to enhance conductivity during scanning. The enamel surface morphology of each group was observed under a scanning electron microscope (SU1510 Scanning Electron Microscope, JEOL, Japan). To reduce observer bias during SEM analysis, the SEM observer was blinded to the experimental group.

## 2.4. Determination of Shear Bond Strength (SBS)

After being stored for 24 hours in water at 37°C, the detached teeth with brackets or attachments were embedded in anhydrite using standardized column molds (20*20*15 mm) with consistent buccal-surface alignment. The embedded teeth were fixed on the electronic universal testing machine (WDW-100 Electronic Universal Testing Machine, Jinan HengRuijin, China) and subjected to shear testing under dry ambient conditions (23 ± 1°C). The dislocation apparatus was positioned perpendicular to the dental surface at the bottom of the bracket and attachment, ensuring that the dislocation force was parallel to the adhesive interface. The dislocation apparatus was loaded at a speed of 1 mm/min until the bracket or attachment detached from the dental surface. An operator blinded to group assignments recorded the peak value of anti-shearing force, and the SBS was calculated (Table 2):

## 2.5. Observation of Fracture Modes of Attachments

The fracture modes of the attachment were quantitatively analyzed using a stereomicroscope with 20 × magnification by two blinded observers, and classified as follows: ① interface fracture (the fracture surface appeared at the bonding interface between the attachment and the enamel), ② mixed fracture (Refers to the simultaneous occurrence of both types of fracture surfaces), ③ cohesive fracture in resin (fracture occured only within the resin attachment), and ④ cohesive fracture in enamel (fracture occured only within the enamel body).

## 2.6. Statistical analysis

Statistical analysis was performed using the SPSS 20.0 (Chicago, IL, USA) program. All data were expressed by means ± standard deviation ($\bar{x} \pm s$). Normality was confirmed by Shapiro-Wilk tests (α = 0.05), and variance homogeneity was verified with Levene's test before parametric analyses. Between-group variation was assessed using a t-test or

**Table 2. The results of the SBS of each group (MPa).**

| Groups | Specific values | | | | | | | | | | SBS (MPa) $\bar{x} \pm s$ | Range |
|--------|------|-------|-------|-------|-------|-------|-------|-------|-------|-------|-----------|-------|
| O1 | 8.78 | 7.28 | 7.12 | 8.61 | 9.50 | 9.72 | 7.19 | 7.36 | 9.32 | 7.51 | 8.24 ± 1.05 | 7.12-9.72 |
| O2 | 12.90 | 12.55 | 9.73 | 11.95 | 10.13 | 12.68 | 12.77 | 10.32 | 13.23 | 11.17 | 11.74 ± 1.30 | 9.73-13.23 |
| A1 | 3.10 | 3.47 | 5.82 | 5.32 | 3.59 | 4.88 | 4.96 | 2.84 | 4.86 | 5.12 | 4.40 ± 1.04 | 2.84-5.82 |
| A2 | 6.95 | 4.37 | 6.45 | 5.26 | 6.72 | 4.67 | 7.08 | 5.38 | 5.06 | 6.85 | 5.88 ± 1.03 | 4.37-7.08 |
| A3 | 7.51 | 4.53 | 5.26 | 7.33 | 5.25 | 6.21 | 4.42 | 7.14 | 5.87 | 5.62 | 5.91 ± 1.12 | 4.42-7.51 |
| B1 | 9.42 | 7.50 | 8.43 | 8.48 | 6.12 | 7.32 | 8.65 | 7.75 | 9.60 | 8.73 | 8.20 ± 1.05 | 6.12-9.60 |
| B2 | 9.98 | 10.15 | 10.50 | 8.90 | 10.05 | 8.48 | 9.62 | 8.12 | 6.30 | 7.97 | 9.01 ± 1.31 | 6.30-10.50 |
| B3 | 10.48 | 9.13 | 11.08 | 9.88 | 8.35 | 9.87 | 10.53 | 9.08 | 10.12 | 8.55 | 9.71 ± 0.90 | 8.35-11.08 |
| C1 | 11.33 | 10.88 | 9.20 | 10.38 | 11.07 | 11.57 | 9.93 | 10.23 | 8.23 | 11.07 | 10.39 ± 1.04 | 8.23-11.57 |
| C2 | 12.47 | 9.87 | 12.32 | 12.35 | 12.65 | 11.95 | 9.30 | 12.93 | 12.32 | 11.18 | 11.73 ± 1.23 | 9.30-12.93 |
| C3 | 12.35 | 10.35 | 13.50 | 13.52 | 9.85 | 11.52 | 9.22 | 12.05 | 12.15 | 13.32 | 11.78 ± 1.54 | 9.22-13.52 |

SBS (MPa) = anti-shearing force (N)/bonding area ($mm^2$).

one-way ANOVA (with Tukey's HSD post-hoc tests for significant effects). Power analysis ($1 - \beta = 0.85$, $\alpha = 0.05$) determined the adequate sample size. All tests were performed at a 95% confidence. A $P$-value≤0.05 was considered significant. ($^{ns}P > 0.05$, $*P < 0.05$, $**P < 0.01$, $***P < 0.001$, $****P < 0.0001$)

## 3. Results

### 3.1. Scanning Electron Microscope (SEM)

Surface of Enamel: Under SEM, the surface of healthy enamel appeared smooth and flat, while that of fluorotic enamel seemed rough, exhibiting varying degrees of pit defects (Fig 1). These structural differences may influence the bonding effectiveness, as the irregular surface can impact the penetration of the adhesive.

Acid Etching Effects: Phosphoric acid etching produced time-dependent structural changes: ① 30s etching: incomplete etching of the enamel surface was observed (Fig 2A); ② 60s etching: a uniform depression around the enamel rod was evident, displaying an irregular honeycomb shape (Fig 2B); this micro-retentive pattern is crucial for resin tag penetration and enhancing bonding strength. ③ 90s etching: the dissolution of the enamel rod along with a smear layer on the enamel surface was visible (Fig 2C), which may compromise mechanical interlocking.

Er: YAG Laser-Assisted Acid Etching: The combined laser-acid treatment produced distinct effects: ① Er: YAG + 30s etching: the enamel surface was pitted and the boundary around the enamel rod was clear (Fig 2D); ② Er: YAG + 60s etching: the enamel surface exhibited a uniform fish scale-like texture, with the center of the enamel rod being concave and the edge evident (Fig 2E); ③ Er: YAG + 90s etching: irregular pits appeared on the enamel surface with enamel dissolution, and the boundary between the enamel rods became unclear (Fig 2F).

### 3.2. Shear Bond Strength (SBS)

The specific values of SBS for each group are listed in Table 2. Group A1 (fluorotic enamel + 30s etching + bracket) showed significantly lower SBS than Group O1 (healthy enamel + 30s etching + bracket) (independent t-test, $P < 0.0001$) (Fig 3A). Similarly, Group B1 (fluorotic enamel + 30s etching + attachment) exhibited markedly reduced strength compared to Group O2 (healthy enamel + 30s etching + attachment) (independent t-test, $P < 0.0001$). Notably, Group O2 (11.74 ± 1.30MPa) and Group C (fluorotic enamel + Er: YAG & acid etching + attachment, 11.30 ± 1.40MPa) demonstrated comparable SBS values (independent t-test, $P = 0.387$) (Fig 3B).

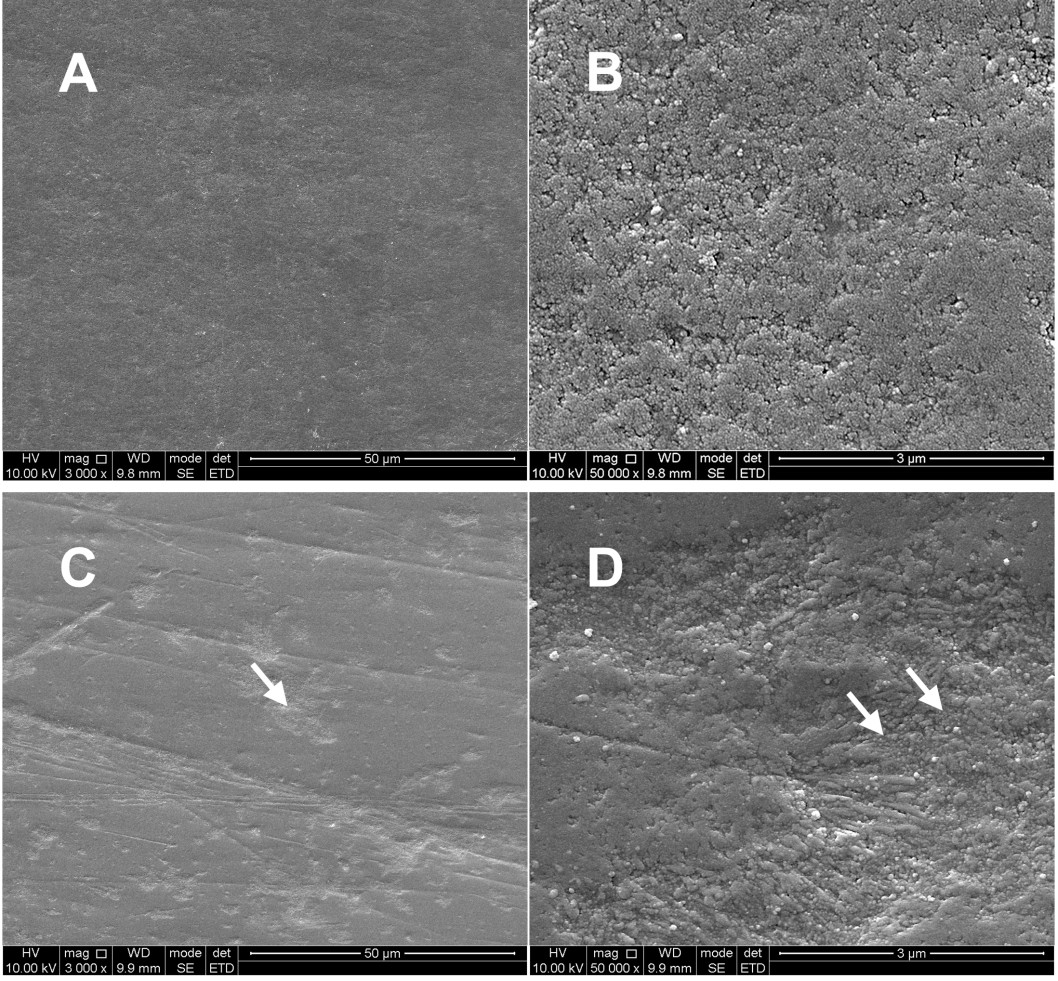

**Fig 1. Micro appearance of the enamel surface.** (a) healthy teeth group: smooth and flat enamel surface (×3000, 50μm); (b) healthy teeth group: smooth and flat enamel surface (×50000, 3μm); (c) dental fluorosis group: rough enamel surface, exhibiting varying degrees of pit defects (×3000, 50μm); (d) dental fluorosis group: rough enamel surface, exhibiting varying degrees of pit defects (×50000, 3μm).

Among fluorotic enamel groups, Group B (acid etching only +attachment) showed significantly higher SBS than Group A (acid etching only +bracket) while showing lower SBS relative to Group C (Er: YAG+etching +attachment) (one-way ANOVA with Tukey's test, $P<0.0001$) (Fig 3C).

There are differences in the overall mean of SBS among groups A, B, and C. Further subgroup analysis revealed no significant differences between: A2 (fluorotic enamel with 60s etching+bracket) vs. A3 (90s etching+bracket); B2 (60s etching+attachment) vs. B1/B3 combinations; C2 (Er: YAG+60s etching+attachment) vs. C3 (Er: YAG+90s etching+attachment)(all $P>0.05$) (Fig 3D).

### 3.3. Fracture modes of attachments

The fracture modes in group B primarily consisted of 40.00% adhesive interface fracture and 46.67% mixed fracture, whereas those in group C were mainly 50% mixed fracture and 33.33% cohesion fracture in resin (Fig 4). This pattern aligns with established mechanical principles where predominant mixed/resin-cohesive fractures (as in Group C) correlate with superior bond strength, indicating effective stress distribution within the adhesive layer.

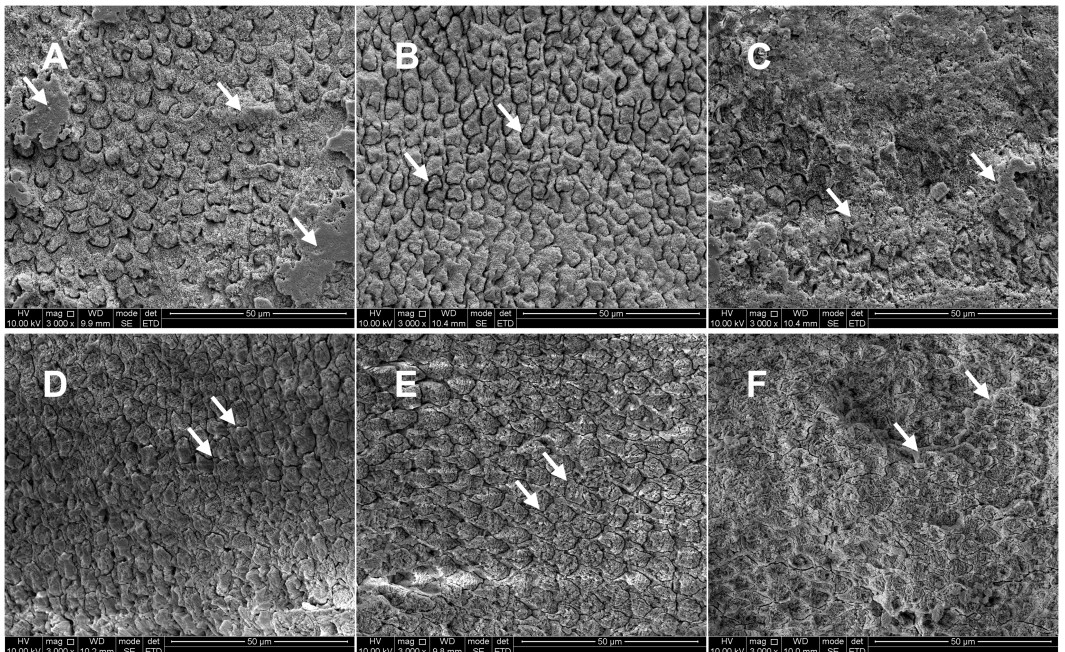

**Fig 2. Micro appearance of the enamel surface of dental fluorosis after different interventions: (a) etching for 30s group: incomplete etching of the enamel surface was observed (arrow); (b) etching for 60s group: a uniform depression around the enamel rod was evident, displaying an irregular honeycomb shape (arrow); (c) etching for 90s group: the dissolution of the enamel rod along with a smear layer on the enamel surface was visible (arrow); (d) etching for 30s after Er: YAG laser irradiating group: the enamel surface was pitted and the boundary around the enamel rod was clear (arrow); (e) etching for 60s after Er: YAG laser irradiating group: the enamel surface exhibited a uniform fish scale-like texture, with the center of the enamel rod being concave and the edge evident (arrow); (f) etching for 90s after Er: YAG laser irradiating group: irregular pits appeared on the enamel surface with enamel dissolution, and the boundary between the enamel rods became unclear (arrow). (all × 3000, 50μm).**

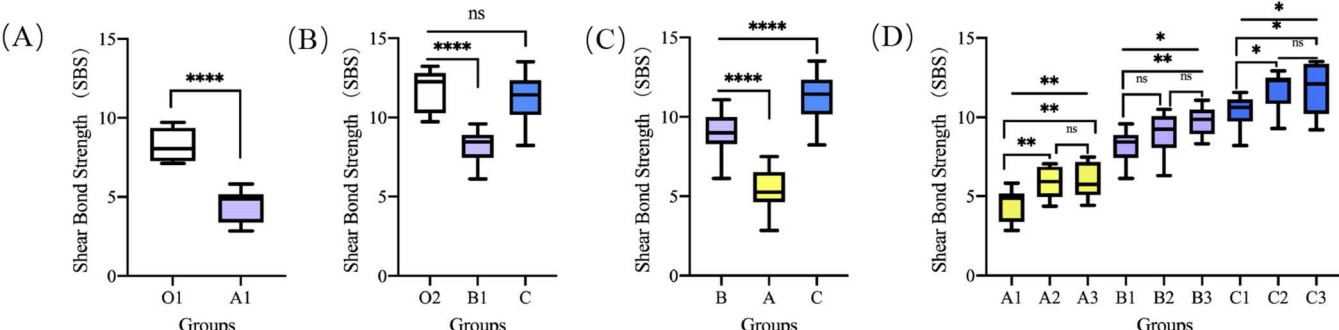

**Fig 3. Box plots showing the SBS between the different groups: (A) SBS comparison between O1 and A1: Group A1 showed significantly lower SBS than Group O1 ($P < 0.0001$); (B) SBS comparison between O2, B1 and A1: Group B1 exhibited markedly reduced strength compared to Group O2 ($P < 0.0001$).** Notably, Group O2 and Group C demonstrated comparable SBS values ($P = 0.387$); (C) SBS comparison between B, A and C: Group B showed significantly higher SBS than Group A while showing lower SBS relative to Group C ($P < 0.0001$); (D) SBS comparison between A1, A2, A3, B1, B2, B3, C1, C2 and C3. (ns$P > 0.05$, *$P < 0.05$, **$P < 0.01$, ***$P < 0.001$, ****$P < 0.0001$).

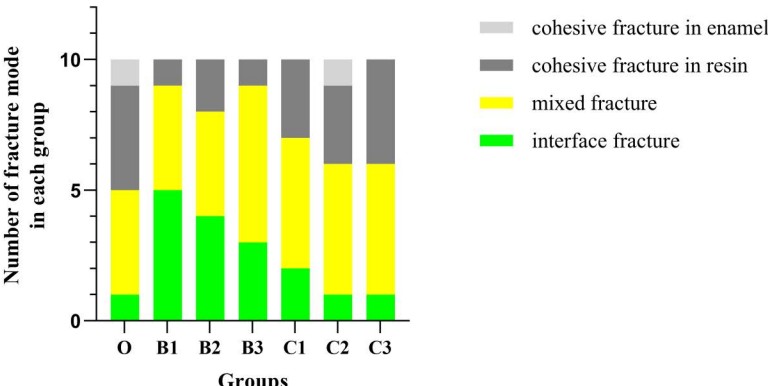

**Fig 4. Bar distribution of fracture patterns in each group.**

## 4. Discussion

Dental fluorosis is generally deemed as one of the common dental diseases, which is caused by successive fluoride exposure during tooth development [16,17]. The more serious the fluorosis, the higher the average fluoride content on the enamel surface [18]. Severity of dental fluorosis would change in an age-related manner and influence the aesthetics of the smile [19]. As more people with dental fluorosis are tending to seek for orthodontic therapy, the debonding of metal brackets from fluorotic enamel challenges the orthodontist [20]. The micro tensile strength of various degrees of dental fluorosis after surface preparation had a significant difference [21]. The etching mode of mild dental fluorosis is similar to that of healthy teeth [20,22], while the bonding performance of severe dental fluorosis is significantly reduced [23].

According to the newly modified Dean index [24], 90 teeth with moderate dental fluorosis were selected for this research to minimize the influence of different enamel surface structures associated with various degrees of dental fluorosis. SEM clearly showed the surface characteristics of moderate fluorotic enamel: fluorotic areas exhibited depressed enamel defects with a groove-like appearance of varying depths. These results approximated findings described in the literature [25,26]. This irregular surface structure results from fluoride deposition on the enamel surface, leading to disordered crystal arrangement, increased microporosity, or surface demineralization. In this study, SBS of both brackets and attachments on healthy teeth was significantly higher than that of the fluorosis group. Bassir et al. found that fluorosis reduced the bonding performance of enamel and had nothing to do with the properties of bonding materials (metal or resin) [27], which was consistent with our hypothesis.

Our study found that the bonding strength of attachments on both healthy enamel and fluorotic enamel was significantly higher than that of the bracket group. According to Gorler et al.'s study [28], the bonding strength of 6–8MPa can meet the clinical needs of the bonding strength of brackets in fixed orthodontics, which is close to the results of our study, 7.12–9.72MPa. The adhesion of the attachment on healthy teeth was 9.73–13.23MPa, which also indicates that the resin attachment has excellent adhesive properties. This may be relevant to the different bonding mechanisms of attachments and brackets on the enamel surface. The attachment was composed of resin, which penetrated the micropores of the enamel surface, forming a resin protrusion that creates the micromechanical locking and retention with the enamel surface [29]. In contrast, the bonding of the bracket to the enamel primarily relied on the chemical interaction between the enamel and the base of the bracket [30].

Compared to conventional 30-second etching, SEM revealed that the enamel interface was irregular and exhibited a honeycomb-like appearance when formed by 60-second etching. The SBS of the attachment increased with an appropriate extension of the acid etching time (60s). However, SEM indicated partial dissolution of the enamel rod and destruction

of the etched interface with 90-second etching. With further extension of acid etching time to 90s, the SBS of attachments no longer increased. Our study demonstrated that prolonging the etching time enhanced the bonding strength of the fluorotic surface to the brackets. According to Silva et al [31], extending etching time can provide proper bonding strength for dental fluorosis and specimens. Due to the decreased mineralization of fluorotic enamel, conventional acid etching (e.g., 37% phosphoric acid) may be ineffective at removing the smear layer, leading to insufficient penetration of adhesives. An appropriate extension of the acid etching time is necessary to meet clinical bonding requirements. According to Torres-Gallegos et al.'s study [32], mild and moderate dental fluorosis can achieve the same bonding effect as normal enamel after acid etching for 30–60 seconds. Excessive acid etching not only damages enamel but also penetrates into dentin, causing dentin demineralization, which affects matrix metalloproteinase (MMP) activity and increases the risk of caries [33].

Group C ($11.30 \pm 1.40$MPa, dental fluorosis + Er: YAG + acid etching) showed approximately the same SBS as healthy teeth ($11.74 \pm 1.30$Mpa, acid etching only), which was higher than that of Group B (dental fluorosis + acid etching only). This indicates that the combination of Er: YAG laser and acid etching is beneficial for improving bonding ability. The results of attachment fracture modes further confirmed the reliability of this research. Nowadays, various lasers are utilized in dental treatments, such as gingivectomy for correcting a gummy smile [34,35], and maxillofacial surgery due to their effectiveness in hemostasis and reducing pain and swelling [36]. As the first laser for dental hard tissue application [37], Er: YAG (2940 nm) demonstrates multifunctional utility in periodontitis, dentin hypersensitivity, and dental restoration [38–41]. Alavi et al.'s study showed that laser etching at 100 mJ energy produced bond strength similar to acid etching [42]. In contrast, Sallam et al. [43] found that there is no significant difference in SBS between Er: YAG laser and acid etching, respectively. Kuhn et al. argue that the Er: YAG laser may cause damage to resin composites, which could limit the use of laser irradiation [44]. However, their study focused on healthy enamel, a significant difference considering the lower reactivity of fluorotic enamel. The Er: YAG laser (2940 nm) is highly absorbed by water and hydroxyapatite in enamel, resulting in the micro-explosion phenomenon. This effect effectively and selectively removes areas of lesions or mineralization abnormalities, resulting in a clean, rough, yet not excessively destructive honeycomb structure on the surface of fluorotic enamel. This not only provides the basis for the micromechanical interlocking structure and increases the bonding area, but also removes the smear layer and exposes a more reactive fresh enamel, facilitating resin penetration and chemical bond formation. The SEM results from group C observed in this study support this mechanism.

Various studies have examined laser etching with different irradiation settings. In this study, the laser parameters were configured according to the device's instructions to ensure the safety and efficacy of fluorosis enamel treatment. We chose a mode specifically designed for dental hard tissue, particularly hypoplastic enamel. The energy output was limited to 100 mJ (30 Hz frequency) to prevent structural damage while effectively altering the enamel surface [45]. A 250 µs ultra-short pulse duration was utilized to significantly reduce the thermal impact on the pulp [46]. Finally, based on SEM results from Group C, a 10-second irradiation produced an ideal honeycomb surface morphology suitable for resin bonding, while extended irradiation times caused structural damage.

The limited sample size was viewed as a limitation of this study, as it may have reduced the statistical power of the subgroup analyses. Another limitation was the lack of oral conditions such as saliva, enzymes, and chewing force. Additionally, only a single laser parameter was tested. Future work should validate the protocols in vivo and investigate Er: YAG parameters for severe fluorosis (TFI = 5–9).

## 5. Conclusion

Within the limitations of this current study, we concluded that:

#1: Fluorotic enamel exhibits lower bond strength than healthy enamel.

#2: Er: YAG laser pretreatment enhances attachment bond strength compared to acid etching alone.

#3: Prolonged acid etching (>60s) degrades enamel rod integrity, thereby reducing bond strength.

For orthodontic practices, we recommend clear aligners as the primary appliance for patients with dental fluorosis and Er: YAG laser-assisted acid etching (60s) for pretreating fluorotic enamel.

## Supporting information

**S1 File. Data-Fracture modes.**
(XLSX)

**S2 File. Data-SBS.**
(XLSX)

**S3 File. Data-fracture modes.**
(PZFX)

## Author contributions

**Data curation:** Rui Xia, Jie Lei.

**Funding acquisition:** Yao Xiao.

**Investigation:** Rui Xia, Jie Lei.

**Methodology:** Rui Xia, Jie Lei.

**Project administration:** Yao Xiao.

**Supervision:** Maoxuan Luo, Yao Xiao.

**Writing – original draft:** Rui Xia, Jie Lei.

**Writing – review & editing:** Maoxuan Luo.

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
