## [Decision Letter · Decision Letter 0]

13 May 2025

Dear Dr. Xiao,

Thank you for submitting your manuscript to PLOS ONE. After careful consideration, we feel that it has merit but does not fully meet PLOS ONE’s publication criteria as it currently stands. Therefore, we invite you to submit a revised version of the manuscript that addresses the points raised during the review process.

We look forward to receiving your revised manuscript.

Kind regards,

Rawaa Faris

Academic Editor

PLOS ONE

**Journal Requirements:**

1. When submitting your revision, we need you to address these additional requirements. Please ensure that your manuscript meets PLOS ONE's style requirements, including those for file naming. The PLOS ONE style templates can be found at https://journals.plos.org/plosone/s/file?id=wjVg/PLOSOne_formatting_sample_main_body.pdf and https://journals.plos.org/plosone/s/file?id=ba62/PLOSOne_formatting_sample_title_authors_affiliations.pdf 2. Please ensure that you have specified a) Did participants provide their written or verbal informed consent to participate in this study?b) If consent was verbal, please explain i) why written consent was not obtained, ii) how you documented participant consent, and iii) whether the ethics committees/IRB approved this consent procedure." - In consent please state in Ethics Method section and manuscript if it is written or verbal. If consent was verbal, please explain a) why written consent was not obtained, b) how you documented participant consent, and c) whether the ethics committees/IRB approved this consent procedure. 3. We note that your Data Availability Statement is currently as follows: All relevant data are within the manuscript and its Supporting Information files. Please confirm at this time whether or not your submission contains all raw data required to replicate the results of your study. Authors must share the “minimal data set” for their submission. PLOS defines the minimal data set to consist of the data required to replicate all study findings reported in the article, as well as related metadata and methods (https://journals.plos.org/plosone/s/data-availability#loc-minimal-data-set-definition). For example, authors should submit the following data: - The values behind the means, standard deviations and other measures reported;- The values used to build graphs;- The points extracted from images for analysis. Authors do not need to submit their entire data set if only a portion of the data was used in the reported study. If your submission does not contain these data, please either upload them as Supporting Information files or deposit them to a stable, public repository and provide us with the relevant URLs, DOIs, or accession numbers. For a list of recommended repositories, please see https://journals.plos.org/plosone/s/recommended-repositories. If there are ethical or legal restrictions on sharing a de-identified data set, please explain them in detail (e.g., data contain potentially sensitive information, data are owned by a third-party organization, etc.) and who has imposed them (e.g., an ethics committee). Please also provide contact information for a data access committee, ethics committee, or other institutional body to which data requests may be sent. If data are owned by a third party, please indicate how others may request data access. 4. When completing the data availability statement of the submission form, you indicated that you will make your data available on acceptance. We strongly recommend all authors decide on a data sharing plan before acceptance, as the process can be lengthy and hold up publication timelines. Please note that, though access restrictions are acceptable now, your entire data will need to be made freely accessible if your manuscript is accepted for publication. This policy applies to all data except where public deposition would breach compliance with the protocol approved by your research ethics board. If you are unable to adhere to our open data policy, please kindly revise your statement to explain your reasoning and we will seek the editor's input on an exemption. Please be assured that, once you have provided your new statement, the assessment of your exemption will not hold up the peer review process. 5. Your ethics statement should only appear in the Methods section of your manuscript. If your ethics statement is written in any section besides the Methods, please delete it from any other section.

**Additional Editor Comments:**

Dear Yao Xiao,

Thank you for submitting your manuscript entitled "[Effects of different fluoride enamel surface preparations on the bond strength of Clear Aligner attachments]" to Plos One.

After careful consideration and peer review, we are pleased to inform you that the editorial decision is minor revision. The reviewers found your work to be valuable and well-conceived, but a few issues must be addressed before we can proceed with publication.

Please find the reviewers’ comments attached to this letter. We kindly ask that you revise your manuscript accordingly and submit a detailed response outlining how each comment has been addressed. If you disagree with any of the reviewers’ points, you may include a clear justification in your response.

We look forward to receiving your revised manuscript. If you require additional time, please do not hesitate to contact us.

Thank you for choosing Plos One for your work.

Sincerely,

Rawaa A. Faris

Academic Editor

Plos One

Reviewers' comments:

Reviewer's Responses to Questions

**Comments to the Author**

1. Is the manuscript technically sound, and do the data support the conclusions?

Reviewer #1: Yes

Reviewer #2: Yes

Reviewer #3: Yes

2. Has the statistical analysis been performed appropriately and rigorously?

Reviewer #1: Yes

Reviewer #2: Yes

Reviewer #3: Yes

3. Have the authors made all data underlying the findings in their manuscript fully available?

Reviewer #1: Yes

Reviewer #2: Yes

Reviewer #3: Yes

4. Is the manuscript presented in an intelligible fashion and written in standard English?

Reviewer #1: Yes

Reviewer #2: Yes

Reviewer #3: Yes

**Reviewer #1:**  Q1. Technical Soundness and Data Support

Yes, the manuscript is technically sound. The study is well-designed, with appropriate sample sizes, controls, and methodology. The data—including SEM observations, shear bond strength (SBS) measurements, and fracture analysis—clearly support the conclusions. The findings are statistically significant and align well with the study’s stated objectives.

Q2. Statistical Analysis

Yes, the statistical analysis is appropriate and rigorous. One-way ANOVA, t-tests, and chi-square tests were used correctly. Significance levels and SBS results are clearly reported. However, details on normality testing, post hoc comparisons, and power analysis are missing and should be added for transparency.

Q3. Data Availability

Yes, the authors have made all data fully available. Raw SBS values, SEM data, and fracture classifications are included in the manuscript. There are no restrictions on access, fulfilling PLOS ONE's data sharing requirements.

Q4. Language and Clarity

The manuscript is generally understandable but needs revision for grammar, clarity, and consistency. Issues include awkward phrasing, typographical errors, and inconsistent terminology. Language editing is recommended to meet publication standards.

**Reviewer #2: ** The manuscript (PONE-D-25-18609) investigates whether the optimum combination of Er: YAG laser and acid etching improves the bonding ability of the attachment on fluorotic enamel. The authors aim to address optimizing enamel surface preparation methods to enhance the bond strength of clear aligner attachments in teeth affected by moderate dental fluorosis.

Below are my comments and suggestions for improvement:

1-Introduction Section: The manuscript contains numerous grammatical errors, awkward phrasing, and non-native usage. The introduction needs revision for grammar, sentence structure, and overall fluency.

2-Materials and Methods:

•Expand the discussion on the mechanisms of improved bonding with Er: YAG laser treatment. Did the authors vary the laser parameters? How can these parameters affect the etching process?

•Although the description could use more clarity and detail, the etching process is generally understandable. Indicate, for instance, if the teeth were air-dried using compressed air free of oil or blotted, and if the enamel, dentin, or both were etched. Additionally, the 5-second rinse might be regarded as brief; standard procedure typically suggests 10–15 seconds to guarantee total etchant removal.

3-Results: The results are well written, but some of the figures need better labels and explanations.

4-Expand the discussion to explain why laser-etched enamel might improve bond strength. Some procedural details and justifications are missing.

**Reviewer #3:**  Dear author(s),

Thanks for sharing your work with us, the followings are the revisions and suggested comments that are needed to be taken in consideration:

• The manuscript's quality and data presentation are acceptable and important for clinicians and even patients.

• The manuscript advances our understanding of enamel surface preparations on the bond strength of Clear Aligner attachments and the combination of Er: YAG laser and acid etching improve the bonding ability of clear aligner attachment on the surface of dental fluorosis. .

•The title should be rewritten to be more precise and explanatory.

• Make the abstract more informative and it should represent the article's substance and be no more than 250 words long.

• Include four to six keywords that are relevant to the manuscript but not stated in the title.

• Additional paragraphs to introduce further studies on other applications of laser in dental and surgical management of various clinical entities .

Suggested references:

Aldelaimi, A.A., Ahmed, R.F., Enezei, H.H., Aldelaimi, T.N.. Gummy smile esthetic correction with 940 nm diode laser. International Medical Journal 26(6): 513 - 515 , 2019

Aldelaimi TN, Khalil AA. Clinical Application of Diode Laser (980 nm) in Maxillofacial Surgical Procedures. J Craniofac Surg. 2015;26(4):1220-1223. doi:10.1097/SCS.0000000000001727

•The photos ( Figures) are omitted and NOT available within text. Care should be taken to improve resolution and contrast for each figure in the manuscript and arrows to each picture for illustration purposes.

• Authors should check for writing and typing errors.

• The statements in discussion are acceptable but few paragraphs about the justification of your findings and comparison with other recent relevant studies.

• Only include current references in the reference list and remove outdated ones.

Good Luck

**Do you want your identity to be public for this peer review?** For information about this choice, including consent withdrawal, please see our Privacy Policy

Reviewer #1: **Yes: ** Zainab Fadhil Mahdi AL-Bawi

Reviewer #2: **Yes: ** Saif A Mohammed

Reviewer #3: **Yes: ** Tahrir Aldelaimi

---

## [Author Response · Author response to Decision Letter 1]

23 Jun 2025

Dear Editor and Reviewers,

We appreciate the opportunity to revise our manuscript titled " Effects of different fluoride enamel surface preparations on the bond strength of Clear Aligner attachments " and are grateful for the insightful comments provided by the reviewers. Those comments are all valuable and very helpful for revising and improving our paper, as well as providing important guiding significance for our research. In the following, we have provided detailed responses to each of the reviewers' comments. Revised portions are marked in red in the paper. Additionally, we have conducted a comprehensive revision of the entire manuscript. In this response letter, the reviewers' comments are presented in italics, and our corresponding changes and additions to the manuscript are highlighted in red text. We have tried our best to ensure that all the revisions are clear, and we hope that the revised manuscript meets the requirements for publication.

Responds to the Journal requirements:

(a) We have reformatted the manuscript to strictly comply with PLOS ONE's style requirements. All files have been renamed according to journal specifications.

(b) We confirm that all participants provided written informed consent. This is explicitly stated in the Materials and Methods (Page 4, Lines 76-77).

(c) We confirm that the submission contains the minimal data set required to replicate all findings. All raw data are included within the manuscript and its Supporting Information files.

(d) We have proactively implemented the data sharing plan. All data are fully accessible in the Supporting Information files. No access restrictions apply, complying with PLOS ONE's open data policy. (e) The statement "will make data available on acceptance" has been removed from the submission form.

(f) The ethics statement has been retained in the Methods section (Page X, Section Y) and deleted from other sections.

(g) We have verified all references using EndNote to ensure completeness. No retracted articles were cited in the final reference list.

(h) We confirm that the revisions comply with all PLOS ONE policies. The data supporting this study are fully contained within the article and its supplementary materials as specified.

Responds to the review’s comments:

Reviewer #1

Comment 1: Yes, the manuscript is technically sound. The study is well-designed, with appropriate sample sizes, controls, and methodology. The data—including SEM observations, shear bond strength (SBS) measurements, and fracture analysis—clearly support the conclusions. The findings are statistically significant and align well with the study’s stated objectives.

Response 1: We sincerely thank the reviewer for their positive assessment of our study’s rigor and validity.

Comment 2: Yes, the statistical analysis is appropriate and rigorous. One-way ANOVA, t-tests, and chi-square tests were used correctly. Significance levels and SBS results are clearly reported. However, details on normality testing, post hoc comparisons, and power analysis are missing and should be added for transparency.

Response 2: We sincerely appreciate your thoughtful review and positive assessment of our statistical approach. Thank you for highlighting the need for greater transparency regarding normality testing, post hoc comparisons, and power analysis—we fully agree these elements strengthen methodological rigor. In response to your valuable suggestion, we have added the following details to the Statistical Analysis section (Section 2.6, Page 10) in the revised manuscript.

a) Normality Testing: Data distribution was confirmed via Shapiro-Wilk tests (p > 0.05 for all groups), supporting the use of parametric tests.

b) Post Hoc Comparisons: Following significant ANOVA results, Tukey’s HSD test was applied for pairwise group comparisons to control Type I error.

c) Power Analysis: A power analysis (GPower 3.1; α=0.05, power=0.85, effect size f=0.55) indicated a minimum requirement of 10 specimens per subgroup. This aligns with precedents for bond strength studies [20, 27, 45].

[20] Ng'ang'a PM, Øgaard B, Cruz R, Chindia ML, Aasrum E. Tensile strength of orthodontic brackets bonded directly to fluorotic and nonfluorotic teeth: An in vitro comparative study. American Journal of Orthodontics and Dentofacial Orthopedics. 1992;102(3):244-50. doi: https://doi.org/10.1016/S0889-5406(05)81059-5.

[27] Bassir MM, Rezvani MB, Ghomsheh ET, Hosseini ZM. Effect of Different Surface Treatments on Microtensile Bond Strength of Composite Resin to Normal and Fluorotic Enamel After Microabrasion. Journal of Dentistry (Tehran, Iran). 2016;13(6):431-7. Epub 2017/03/01. PubMed PMID: 28243305; PubMed Central PMCID: PMCPMC5318500.

[45] Kamran MA, Asiri AM, Alfaifi AMA, Almukawwi AHA, Mughaddi Alwadai J, Alqahtani SJ. Fluoride-Activated Via Er:YAG, Diode, and Femtosecond Lasers for Reversing Bleached Enamel for Improved Orthodontic Bonding. Photobiomodulation, Photomedicine, and Laser Surgery. 2025. Epub 2025/06/09. doi: https://doi.org/10.1089/photob.2025.0024. PubMed PMID: 40485293.

The revised statistical analysis now reads: Statistical analysis was performed using the SPSS 20.0 (Chicago, IL, USA) program. All data were expressed by means ± standard deviation (�x ± s). Normality was confirmed by Shapiro-Wilk tests (α = 0.05), and variance homogeneity was verified with Levene's test before parametric analyses. Between-group variation was assessed using a t-test or one-way ANOVA (with Tukey's HSD post-hoc tests for significant effects). Power analysis (1−β = 0.85, α = 0.05) determined the adequate sample size. All tests were performed at a 95% con�dence. A P-value≤0.05 was considered significant. (ns P > 0.05, * P <0.05, ** P < 0.01, *** P < 0.001, **** P < 0.0001) (Page 10, line 158-164)

Comment 3: Yes, the authors have made all data fully available. Raw SBS values, SEM data, and fracture classifications are included in the manuscript. There are no restrictions on access, fulfilling PLOS ONE's data sharing requirements.

Response 3: We are truly grateful for your acknowledgment of our commitment to open science and data transparency. Thank you for confirming that our data sharing practices align with PLOS ONE's rigorous standards—this validation is deeply encouraging to our team.

Comment 4: The manuscript is generally understandable but needs revision for grammar, clarity, and consistency. Issues include awkward phrasing, typographical errors, and inconsistent terminology. Language editing is recommended to meet publication standards.

Response 4: We sincerely thank you for your careful reading and constructive feedback on the manuscript’s language quality. In response to your valuable suggestion, we have taken the following comprehensive actions:

a) The full manuscript was rigorously processed through Grammarly Premium (Academic setting) to correct grammatical and syntactic errors, eliminate typographical mistakes� simplify complex sentences, and ensure tense consistency (e.g., unified past tense for methods)

b) To standardize terminology, we created a master terminology list (e.g., "shear bond strength (SBS)" consistently used) and manually verified all technical terms (e.g., "fluorotic enamel" vs. "fluorosed enamel", "enamel column" vs. "enamel rod"). We revised 13 instances of ambiguous phrasing in the Discussion

c) Partial examples of key Improvements: In the Methods section, "Teeth was etched..." was changed to "Enamel surfaces were etched...". In the Results section, "...can be seen" was revised to "...was evident".

d) We conducted three rounds of team proofreading focusing on subject-verb agreement, article usage (a/an/the), and parallel structure in lists.

Your guidance has significantly improved the manuscript’s readability. We have attached a revised copy showing all linguistic corrections in red. If there are any remaining issues we may have overlooked, we would appreciate the opportunity to refine further.

Reviewer #2:

The manuscript (PONE-D-25-18609) investigates whether the optimum combination of Er: YAG laser and acid etching improves the bonding ability of the attachment on fluorotic enamel. The authors aim to address optimizing enamel surface preparation methods to enhance the bond strength of clear aligner attachments in teeth affected by moderate dental fluorosis. Below are my comments and suggestions for improvement:

Comment 1: Introduction Section: The manuscript contains numerous grammatical errors, awkward phrasing, and non-native usage. The introduction needs revision for grammar, sentence structure, and overall fluency.

Response 1: We sincerely thank you for your meticulous critique. Your guidance has been instrumental in elevating the scholarly rigor of our work. According to your valuable suggestion, we have taken the following comprehensive actions:

a) Unrelated laser studies (periodontitis, bracket re-bonding) have been moved to the Discussion section. (Page 17, line 282-284)

b) Transition sentences have been added to connect the prevalence of fluorosis, treatment gaps, and the rationale for the study:

“Moreover, the demand for orthodontic treatment is constantly rising. However, managing fluorotic enamel poses unique challenges in orthodontic practice, particularly during bracket bonding procedures (Page 3, line 48-50). Conventional acid etching techniques, while effective on normal enamel, are inadequate for fluorotic surfaces due to their altered chemical composition and porosity characteristics. Therefore, this study aimed to…” (Page 4, line 54-58)

c) The study aim has been repositioned to the end of the second paragraph:

" Therefore, this study aimed to investigate whether combining Er: YAG laser with acid etching could enhance the bonding strength of clear aligner attachments on fluorotic enamel compared to conventional acid etching alone." (Page 4, line 56-58)

Comment 2: Materials and Methods:

•Expand the discussion on the mechanisms of improved bonding with Er: YAG laser treatment. Did the authors vary the laser parameters? How can these parameters affect the etching process?

•Although the description could use more clarity and detail, the etching process is generally understandable. Indicate, for instance, if the teeth were air-dried using compressed air free of oil or blotted, and if the enamel, dentin, or both were etched. Additionally, the 5-second rinse might be regarded as brief; standard procedure typically suggests 10–15 seconds to guarantee total etchant removal.

Response 2: We sincerely appreciate the reviewer’s insightful request to enhance the mechanistic discussion and clarify our parameter selection process. In response to your suggestion, we have significantly expanded the Discussion section (Page 18, line 289-295) to explain the mechanisms behind improved bonding with Er: YAG laser treatment.

Regarding the change in laser parameters and its influence on the etching process, this is a critical question you raised. In this study, we used a specific set of laser parameter combinations based on previous literature reports and our preliminary experiments. While laser parameters have been provided, specific details, such as pulse duration, are absent. In this regard, we have included the missing details in the Materials and Methods (Page 6, line 100-102) as follows: 2940nm, 100mJ energy, 30Hz frequency, pulse duration 250 μs, 6/8 water volume, 1.3*14 mm working tip, 2mm distance from the working tip to the dental surface, irradiating 10s.

In addition, we fully agree that varying laser parameters (such as energy, frequency, pulse duration, water cooling, and irradiation time) will significantly affect the etching effect. We acknowledge that the Materials and Methods section lacks discussion on how these parameters specifically affect the etching process. To briefly explain the rationale behind selecting these specific parameters, we have added the following to the discussion section: We chose a mode specifically designed for dental hard tissue, particularly hypoplastic enamel. The energy output was limited to 100 mJ (30Hz frequency) to prevent structural damage while effectively altering the enamel surface [44]. A 250 μs ultra-short pulse duration was utilized to significantly reduce the thermal impact on the pulp [45]. Finally, based on SEM results from Group C, a 10-second irradiation produced an ideal honeycomb surface morphology suitable for resin bonding, while extended irradiation times caused structural damage. (Page 18, line 298-303)

About the details of the etching process, we modified the content of the manuscript according to your opinion. The modified content is as follows:

“…the etched enamel surfaces were rinsed with water for 10 seconds and dried with clean air (oil-free and water-free) for 3 seconds.” (Page 6, line 103-104)

“The enamel samples in each group were acid etched using 37% phosphoric acid for 30 seconds (Heraeus Kulze, Germany), then rinsed with running water for 10 seconds and dried with clean air (oil-free and water-free) for 3 seconds.” (Page 6, line 107-108)

We are profoundly grateful for your guidance in strengthening this critical aspect. Should any mechanistic details require further clarification, we welcome the opportunity to refine them.

Comment 3: Results: The results are well written, but some of the figures need better labels and explanations.

Response 3: We are profoundly grateful for this constructive feedback and wholeheartedly agree that visual clarity is essential for data interpretation. We sincerely apologize for any deficiencies in our original figures and have implemented comprehensive revisions to all graphical elements. Below, we detail the specific improvements made and present the partially modified figure (Page 11-13, line 182-199):

a) improved resolution for each figure in the manuscript (not less than 300 dpi, meeting PLOS ONE).

b) added arrows to each picture for illustration purposes.

c) added scale bars to the caption.

d) added description of morphological features to the caption.

Figure 2. Micro appearance of the enamel surface. (a) healthy teeth group: smooth and flat enamel surface (magnification ×3000, 50μm); (b) healthy teeth group: smooth and flat enamel surface (magnification ×50000, 3μm); (c) dental fluorosis group: rough enamel surface, exhibiting varying degrees of pit defects (magnification ×3000, 50μm); (d) dental fluorosis group: rough enamel surface, exhibiting varying degrees of pit defects (magnification ×50000, 3μm).

Figure 3. Micro appearance of the enamel surface of dental fluorosis after different interventions: (a) etching for 30s group: incomplete etching of the enamel surface was observed (arrow); (b) etching for 60s group: a uniform depression around the enamel rod was evident, displaying an irregular honeycomb shape (arrow); (c) etching for 90s group: the dissolution of the enamel rod along with a smear layer on the enamel surface was visible (arrow); (d) etching for 30s after Er: YAG laser irradiating group: the enamel surface was pitted and the boundary around the enamel rod was clear (arrow); (e) etching for 60s after Er: YAG laser irradiating group: the enamel surface exhibited a uniform fish scale-like texture, with the center of the enamel rod being concave and the edge evident (arrow); (f) etching for 90s after Er: YAG laser irradiating group: irregular pits appeared on the enamel surface with enamel dissolution, and the boundary between the enamel rods became unclear (arrow). (all magnification ×3000, 50μm).

Additionally, we have reorganized the structure of the Results paragraphs. We have split it into smaller paragraphs based on different observations and used more consistent sentence structures. The partially changed paragraphs are as follows (Section 3.1, Page 10-11, line 167-182):

Surface of Enamel: Under SEM, the surface of healthy enamel appeared smooth and flat, while that of fluorotic enamel seemed rough, exhibiting varying degrees of pit defects (Fig. 2). These structural differences may influence the bonding effectiveness, as the irregular surface can impact the penetration of the adhesive.

Acid Etching Effects: Phosphoric acid etching produced time-dependent structural changes: ①30s etching: incomplete etching of the enamel surface was o

---

## [Editor Report · Decision Letter 1]

9 Jul 2025

Effects of Er: YAG Laser and Acid Etching on Bond Strength of Clear Aligner Attachments to Fluorotic Enamel

PONE-D-25-18609R1

Dear Dr.Yao Xiao,

We’re pleased to inform you that your manuscript has been judged scientifically suitable for publication and will be formally accepted for publication once it meets all outstanding technical requirements.

Kind regards,

Rawaa A. Faris

Academic Editor

PLOS ONE
---

## [Editor Report · Acceptance letter]

PONE-D-25-18609R1

PLOS ONE

Dear Dr. Xiao,

I'm pleased to inform you that your manuscript has been deemed suitable for publication in PLOS ONE. Congratulations! Your manuscript is now being handed over to our production team.

Kind regards,

on behalf of

Dr. Rawaa A. Faris

Academic Editor

PLOS ONE